# Evaluation of Actual Ventilation Rates and Efficiency in Research-Scale Pig Houses Based on Ventilation Configurations

**DOI:** 10.3390/ani13152451

**Published:** 2023-07-29

**Authors:** Se-yeon Lee, Lak-yeong Choi, Jinseon Park, Kehinde Favour Daniel, Se-woon Hong, Kyeongseok Kwon, Okhwa Hwang

**Affiliations:** 1Department of Rural and Bio-Systems Engineering, Chonnam National University, Gwangju 61186, Republic of Korea; seyeonn@jnu.ac.kr (S.-y.L.); cly6847@jnu.ac.kr (L.-y.C.); kenniedee@jnu.ac.kr (K.F.D.); 2Education and Research Unit for Climate-Smart Reclaimed-Tideland Agriculture, Chonnam National University, Gwangju 61186, Republic of Korea; icarus381@jnu.ac.kr; 3AgriBio Institute of Climate Change Management, Chonnam National University, Gwangju 61186, Republic of Korea; 4Animal Environment Division, Department of Animal Biotechnology and Environment, National Institute of Animal Science, Wanju-gun 55365, Republic of Korea; kskwon0512@korea.kr (K.K.); hoh1027@korea.kr (O.H.)

**Keywords:** air change per hour (ACH), infiltration, tracer gas decay method, ventilation rate, ventilation effectiveness

## Abstract

**Simple Summary:**

This study evaluated the ventilation rates and efficiency in research-scale pig houses by examining different inlet and exhaust configurations. The goal was to understand how well the ventilation system worked. Two pig rooms were studied, and actual ventilation rates were measured using specialized equipment. The results revealed that certain combinations of inlets and exhausts performed better in terms of ventilation rates. However, the measured rates were lower than expected. This study also found that the side exhaust fan closer to where the pigs were active was more effective at providing fresh air compared to the chimney exhaust fan. Additionally, the ceiling inlet provided better air distribution. This study highlights the importance of choosing the right ventilation setup to ensure a healthy environment for both pigs and workers. By improving ventilation efficiency, this research can contribute to creating better conditions for pig farming, resulting in improved animal welfare and productivity.

**Abstract:**

Accurate ventilation control is crucial for maintaining a healthy and productive environment in research-specialized pig facilities. This study aimed to evaluate actual ventilation rates and ventilation efficiency by investigating different inlet and exhaust configurations. The research was conducted in two pig rooms, namely pig room A and pig room B, in the absence of animals and workers to focus solely on evaluating the ventilation system’s performance. Actual ventilation rates were measured using hood-type anemometers, and the local air change per hour was analyzed at various measurement points via tracer gas decay experiments. The results demonstrated that specific inlet and exhaust combinations, such as side inlet/chimney outlet and ceiling inlet/side outlet, exhibited higher ventilation rates. However, the measured ventilation rates were much lower than the manufacturer’s specifications. The side exhaust fan closer to the pig activity space demonstrated better ventilation effectiveness for the animals than the chimney exhaust fan. Additionally, the ceiling inlet exhibited superior air distribution and uniformity. Lower ventilation rates and higher infiltration ratios resulted in reduced ventilation efficiency, with the difference between pig and worker activity spaces being pronounced. This study emphasizes the importance of selecting optimal inlet and exhaust configurations to achieve efficient ventilation and create a healthy environment for both pigs and workers.

## 1. Introduction

Livestock farming is pivotal in meeting the growing demand for animal products worldwide. However, it also presents significant challenges, including the management of air quality within confined animal housing systems. In particular, proper ventilation helps control air temperature, humidity, and gas concentrations, while also reducing the accumulation of airborne pollutants, such as ammonia, odorous compounds, and particulate matter [1]. Adequate ventilation is also essential to ensuring a healthy and comfortable environment for both the animals and the workers involved in farming operations [2,3].

With the advancement of animal husbandry technology, environmental concerns in the livestock industry, such as odor and particulate matter, have become a critical research focus [4]. To address these challenges, the National Institute of Animal Science (NIAS) in Korea has recently constructed a research-scale pig house specifically designed for conducting various ventilation-related studies in this field. The research pig house can offer significant benefits at both research and industry levels. It allows controlled experiments, leading to scientific advancements in livestock management and animal welfare. Optimized ventilation strategies can enhance animal comfort and productivity, guiding the industry to implement best practices. Such research prevents costly mistakes, meets regulatory requirements, and supports sustainable growth in the pig industry.

However, due to inadequate studies that comprehensively evaluate the performance and efficiency of various ventilation configurations in research-scale pig houses, it is difficult to understand how different configurations could impact ventilation rates. There is a need for profound knowledge of how the dispersion of fresh air within the facility can improve the design and analysis of experiments that are conducted in this facility.

Accurately measuring ventilation rates is important when examining ventilation efficiency. The ventilation rate is commonly controlled by the fan operation rate; however, in most situations, fans provide less airflow compared to the prescribed rate by the manufacturer or theoretical values due to aging, the presence of shutters, and nonuniform indoor static pressure [4,5]. To measure the actual ventilation rates, various experimental methods can be employed, such as the use of anemometers or air velocity sensors to directly measure the airflow at specific inlet and outlet points [6]. These instruments can provide real-time airflow velocity data, which can be combined with the cross-sectional area of openings to calculate the volumetric airflow rate. However, determining the best measurement points that represent the total airflow rate is challenging due to the non-uniformity of the velocity profile within the opening [6].

Another method involves using a combination of flow hoods and the multipoint averaging pitot tube. Flow hoods are employed to measure the airflow rate at specific air outlets, making the airflow homogeneous within the cross-section of the flow hoods. Then, the sensing points of the pitot tube are distributed across the cross-sectional area of the airflow path. If the size of the flow hood is nearly identical to that of the exhaust fan is measured and the exhaust fan is properly positioned at the center of the hood, the ventilation rate can be accurately measured with an error of less than 5% [7].

While measuring the ventilation rate is important, it is essential to recognize that the ventilation rate alone may not accurately represent ventilation effectiveness [8]. Ventilation effectiveness concerns the quality and supply of air distribution [9]. In order to accurately determine ventilation effectiveness, several ventilation effectiveness indices must be considered, such as the air diffusion performance index [10], air change effectiveness based on the concept of air’s age [8,10,11,12,13], temperature effectiveness [10,12], pollutant removal efficiency [11,12], the percentage of outside air [11], and draught rating [12]. In addition, air changes per hour (ACH) or the air exchange rate is a common index that denotes the number of times the specific air volume within a livestock building is replaced with fresh air in one hour [14,15,16,17,18]. The ACH can be typically measured through tracer gas decay experiments. The decay rate of the tracer gas can be influenced by the local ventilation rate and air mixing within a livestock building; faster decay indicates a higher ACH. The accuracy of the ACH measurement tends to increase as the ventilation rate increases [15].

In this study, the primary objective was to assess the actual ventilation rates and ventilation effectiveness in research-scale pig houses. Field measurements were conducted in the absence of animals and workers to determine the actual ventilation rates associated with different combinations of three inlet types and two outlet types. By analyzing the variations and distribution in the inflow rates across multiple inlets, we aimed to obtain insights into ventilation characteristics in relation to the combinations of inlet and outlet configurations. Tracer gas experiments were conducted to further analyze the distribution of ventilation effectiveness. The comprehensive analysis conducted in this study aimed to establish and provide information on the ventilation performance according to different ventilation configurations in each pig room, enabling the utilization of these research facilities for future experiments and research in research-scale pig houses for the development of the swine industry.

## 2. Materials and Methods

### 2.1. Experimental Pig Houses

The NIAS constructed two research-specialized pig houses that enable various ventilation-related studies. The pig houses consist of a corridor and multiple experimental pig rooms along the corridor. One pig room was selected from each house, resulting in a total of two pig rooms used for the experiments, as shown in Figure 1. They were referred to as pig room A and pig room B throughout the experiments. The experiments were conducted in the absence of animals. Prior to the experiments, the manure inside the pits was discharged and cleaned to facilitate the use of tracer gases.

The two pig rooms follow a typical corridor-pig room configuration. The entire floor in pig room A was covered with concrete slats, whereas pig room B had a partially slatted floor design, with half of the floor consisting of slats while the remaining portion was made of concrete. The two pig rooms had the same dimensions, with a floor area of 7.8 m × 6.3 m and a height of 3 m. These rooms were designed to accommodate either 164 nursery pigs or 61 finishing pigs. In addition, both pig rooms were equipped with four pens. They also received fresh air from the corridor through openings located on the walls and ceiling.

### 2.2. Ventilation Configurations

Both pig rooms are equipped with three types of air inlets, namely the side wall baffle, ceiling baffle, and duct-type inlets. The baffle inlets are spring-loaded structures that automatically open or close based on the indoor negative pressure. Four baffles were installed on the side walls, whereas eight baffles were installed on the ceiling. The duct-type inlet was designed to supply fresh air from the corridor to the pig rooms through a centrally located flexible duct. Each inlet system could be individually adjusted for operation and control.

The exhaust system of pig room A consisted of exhaust fans installed on the side wall and the pit. In pig room B, there was an exhaust fan on the side wall and a chimney fan installed on the ceiling. Both pig rooms had two exhaust systems installed, allowing for the operation of either one or both systems simultaneously. Taken together, both pig rooms were equipped with three inlet systems and two exhaust systems. If only one type of inlet and one type of exhaust system were operated in each pig room, a total of six ventilation combinations could be obtained for each room.

#### 2.2.1. Air Inlet Units

The side wall inlets were composed of four baffle systems. Each baffle measured 400 mm × 162 mm, and they were designed to open inward into the interior space based on the negative pressure created by the operation of the exhaust fans. On the corridor side, hoods were attached to minimize the impact of the airflow from the corridor on the inflow of fresh air through the baffles (Figure 2).

The ceiling inlets were evenly distributed across eight locations in the pig room, with a total of two inlets per pen. Each inlet was equipped with a baffle system that could open in both directions based on negative pressure conditions (Figure 3). The size of each inlet was 540 mm × 120 mm.

The duct inlet had a total of 24 holes, each of which had a 10 cm diameter, which was distributed at 12 different distances to supply air into the rooms in both directions (Figure 3). The duct was connected to the corridor and allowed fresh air from the corridor to enter the indoor space through 350 mm diameter openings in contact with the corridor.

#### 2.2.2. Exhaust Fan Units

Three types of exhaust fans were used in the experimental pig rooms. The side exhaust fan had an inner diameter of 645 mm, an outer diameter of 773 mm, and a maximum airflow capacity of 11,760 m^3^ h^−1^, making it the largest fan (COCO-630A, Dongsung Cocofan Co. Ltd., Hwaseong-si, Gyeonggi-do, Republic of Korea). A hood bent 90° downward was attached to the outer side of the side exhaust fan for protection against snow, ice, rain, and wind. The chimney exhaust fan had an inner diameter of 540 mm, an outer diameter of 620 mm, and a maximum airflow capacity of 8340 m^3^ h^−1^ (COCO-500A, Dongsung Cocofan Co. Ltd., Hwaseong-si, Gyeonggi-do, Republic of Korea). The pit exhaust fan, being the smallest fan, had an inner diameter of 358 mm, an outer diameter of 452 mm, and a maximum airflow capacity of 4180 m^3^ h^−1^ (COCO-350A, Dongsung Cocofan Co. Ltd., Hwaseong-si, Gyeonggi-do, Republic of Korea).

### 2.3. Measurements

#### 2.3.1. Ventilation Airflow Rates

The actual ventilation rates of exhaust fans can vary depending on the selection of the inlet systems, as it affects the magnitude and distribution of indoor negative pressure. The ventilation rate of the smallest pit exhaust fan was measured using a portable hood-type anemometer (Testo 420, Testo SE & Co., Titisee-Neustadt, Germany) attached to the outer side of the pit exhaust fan (Figure 4). The ventilation rates of the relatively larger side exhaust fan and chimney exhaust fan were measured using large hood-type anemometers developed by NIAS, following the same measurement principle as that of the portable hood-type anemometer (Figure 4). Due to the presence of hoods attached to the outer side of the exhaust fans, it was not possible to install the hood-type anemometer for the side exhaust fan outdoors. Similarly, as a result of structural limitations, the hood-type anemometer could not be installed for the chimney exhaust fan outdoors. Therefore, large hood-type anemometers were attached to the inner side of the two exhaust fans within the indoor space to measure the actual ventilation rates.

The actual ventilation rates of the exhaust fans were measured at fan operation rates of 10, 20, 30, 40, 50, 60, 70, 80, 90, and 100%. These measurements were conducted for each of the three inlet configurations. The baffle inlets were automatically opened by the negative pressure generated by the exhaust fan. Each measurement was performed for 30 s, and three measurements were performed in total. This process was repeated three times to ensure the accuracy and reliability of the measurements.

The inflow rates of the inlet systems were measured to compare the differences in inflow rates based on the inlet locations. The inflow rates were calculated by measuring the air velocity at the inlets using portable thermal anemometers (Testo 405i, Testo SE & Co., Titisee-Neustadt, Germany) and multiplying it by the inlet area. The measurement frequency and duration were consistent with the ventilation rate measurements. Furthermore, the infiltration rate was estimated by analyzing the difference between the total inflow rate and the ventilation rate.

The measurement positions are presented in Figure 1. For the side wall inlets, one anemometer was attached to each corridor-side inlet of the four baffle systems to measure the inflow rates. Similarly, for the ceiling inlets, eight anemometers were installed, with one anemometer for each baffle system. As for the duct inlet system, one anemometer was installed at the corridor-side duct inlet to measure the total inflow rate through the duct. In addition, anemometers were installed at seven holes, with one anemometer installed for every two holes and an extra one very close to the exhaust fan to measure the differences in inflow rates at those positions. Notably, even though the duct had a total of 24 holes distributed on both sides, the measurement was performed only on one side, assuming symmetry between the two directions. In all measurements, the anemometers were installed at the center of the inlet areas.

#### 2.3.2. Tracer Gas Decay Method

The tracer gas decay method was employed to determine the ACH at specific locations within the pig rooms. Carbon dioxide (CO_2_) was used as the tracer gas, and its concentration was measured using 14 portable gas analyzers. Two types of gas analyzers were used: 9 MultiRAE gas detectors (Honeywell International Inc., Charlotte, North Carolina, USA) and 5 GasTiger2000 (Wandi, Shandong, China). Prior to the experiment, all gas analyzers were calibrated to ensure consistent measurement results.

CO_2_ gas was injected into the pig rooms at six locations: four points at the corners of the ceiling and two points at the middle of both sides of the floor. To ensure the uniform dispersion of CO_2_ gas within the room, four mixing fans were installed and operated during gas injection. According to ISO 16000-8 standards [19], the injection tube should be positioned behind the mixing fans for better gas mixing. Therefore, the mixing fans were positioned in front of the injection points on the floor.

CO_2_ gas concentrations were measured at nine evenly selected points in a 3 × 3 grid pattern within the pig room space (Figure 5). The gas concentration changes were measured at a height of 0.3 m above the floor, which corresponds to the respiratory height of the pigs. Additionally, gas concentration changes were measured at a height of 1.5 m, which corresponds to a worker’s respiratory height at five points in a cross-shaped pattern.

The specific steps of the experiment were as follows:(1)Start injecting CO_2_ gas and activate the mixing fans;(2)Stop the gas injection when the concentration of all gas analyzers reaches or exceeds 4500 ppm (the actual concentration is approximately 5000 ppm);(3)Stop the operation of the mixing fans when the concentration difference between the 14 gas analyzers at each point decreases to 10% (500 ppm). This indicates a uniform CO_2_ concentration throughout the pig room;(4)Allow time for the air currents to stabilize;(5)Perform the designated ventilation and stop the measurements when the concentration at all points reaches 1000 ppm.

The ACH was calculated using Equation (1):(1)$$ACH=\frac{1}{t_{2}-t_{1}}\ln\left(\frac{C_{1}}{C_{2}}\right)$$
where ACH is the air changes per hour (h^−1^); *C*_1_ and *C*_2_ represent the concentrations of CO_2_, which are 4000 ppm and 1000 ppm, respectively; and *t*_1_ and *t*_2_ indicate the times in seconds when the CO_2_ concentrations are 4000 ppm and 1000 ppm, respectively.

The variations in CO_2_ concentrations at different locations are influenced by the ventilation process, and these spatial differences represent ventilation efficiency, which is quantified using ACH according to Equation (1). However, during the initial phase, when ventilation starts, the CO_2_ concentration reduction pattern at each location is unstable due to the sudden opening of inlets and the exhaust fans starting, leading to unstable indoor airflow. Considering this unstable period can lead to the overestimation or underestimation of ventilation effectiveness [13,20]. Therefore, we excluded this initial unstable period and defined the starting point (*t*_1_) as the time when the CO_2_ concentration at each location reached 4000 ppm [13]. Consequently, the results from this study are relevant for cases where ventilation operation is sustained for a long period; however, they are inappropriate for scenarios where the ventilation system operates intermittently for a few seconds, such as during the winter season.

### 2.4. Experimental Design and Analysis

The actual ventilation rates were measured for the six ventilation configurations, which were obtained by combining three inlet conditions and two exhaust conditions for each of the two pig rooms (Table 1). For each ventilation configuration, nine fan operation rates were considered. Additionally, the inflow rate at each inlet was measured to analyze the spatial variations in the inflow rates of individual inlets and the potential infiltration rate of the pig rooms.

The measurement of ACH at different positions within the pig rooms using the tracer gas decay method was performed for the same six ventilation configurations as the ventilation rate measurements. However, for pig room A, the ventilation rate was found to be very low when using the pit exhaust fan, and the influence of the inlet conditions was found to be insignificant (refer to Section 3.1). Therefore, only the three ventilation configurations for the side exhaust fan were considered for the tracer gas experiment. Due to the time and cost constraints associated with the tracer gas experiment, measurements were only conducted for two fan operation rates which were adjusted to achieve the ventilation rates of 2000 m^3^ h^−1^ and 5000 m^3^ h^−1^ based on the results of the ventilation rate measurements. The two ventilation rates were selected to represent relatively high and low ventilation rates considering the seasonal change and ventilation requirement.

The experimental periods and weather conditions for each pig room are provided in Table 2. During the experimental periods, no significant differences in air temperature and humidity between the indoor and outdoor environments were observed. In particular, when measuring the ACH at different positions using the tracer gas method, the impact of air temperature differences between the indoor and outdoor environments on buoyancy and indoor airflow was expected to be minimal.

## 3. Results

### 3.1. Ventilation Rates through Exhaust Fans and Inlet Openings

#### 3.1.1. Actual Ventilation Rate

The measured ventilation rates for each combination of the six inlet and outlet configurations, along with the different fan operation rates, are presented in Figure 6. In pig room A, the maximum measured ventilation rates were 6109 m^3^ h^−1^ and 1405 m^3^ h^−1^ for the side exhaust and pit exhaust fans, respectively. In pig room B, the maximum measured ventilation rates were 5402 m^3^ h^−1^ and 7710 m^3^ h^−1^ for the side exhaust and chimney exhaust fans, respectively.

Considering the manufacturer’s provided ventilation rates under no load conditions, which were 11,760 m^3^ h^−1^, 8340 m^3^ h^−1^, and 4180 m^3^ h^−1^ for the side, chimney, and pit exhaust fans, respectively, the actual ventilation rates achieved were only 51.9%, 92.4%, and 33.6% of the maximum capacity, respectively. Among the three fans, only the chimney exhaust fan exhibited a performance close to its original specifications, whereas the side and pit exhaust fans exhibited significant performance degradation. These findings highlight notable performance discrepancies between the manufacturer’s specifications and the actual ventilation rates achieved in the pig rooms.

The performance degradation of the ”Ide exhaust fan could be attributed to the presence of a hood attached to the external side of the fan, which significantly reduced its efficiency. As for the pit exhaust fan, the airflow that entered through the upper inlets above the floor had to pass through the holes in the slatted floor before being discharged by the pit exhaust fan located beneath the floor. This process of passing through the slatted floor introduced considerable resistance and pressure losses, leading to a decrease in fan performance.

The difference in ventilation rates among different inlet configurations was not significant. However, this merely indicates that the three inlets imposed similar resistance on the exhaust fan. As mentioned in Section 3.2, the ventilation effectiveness varied among the three inlets. Overall, the ceiling inlet exhibited higher ventilation rates, whereas the duct inlet exhibited slightly lower ventilation rates. This could be attributed to the total inlet area. The ceiling inlet has a larger number of inlets and the largest total inlet area, whereas the duct inlet had the highest number of individual holes but smaller hole sizes, resulting in the smallest total inlet area.

Regarding fan operation rates, higher fan operation rates unsurprisingly resulted in higher ventilation rates. The relationship between the fan operation rates and ventilation rates presented a gradual S-shaped curve. The most significant changes in ventilation rates were observed when the fan operation rates ranged from 40% to 70%. Beyond an 80% fan operation rate, the increase in ventilation rates became less pronounced.

#### 3.1.2. Inflow Rates through Inlet Openings

The individual inflow rates at each inlet opening or baffle and the total inflow rate of the pig room were measured simultaneously during the ventilation rate measurement. The results for the side, chimney, and duct inlets are presented in Figure 7, Figure 8 and Figure 9, respectively.

The total inflow rate through the side inlet ranged from 1716 to 2584 m^3^ h^−1^, except for the very low value of 128 m^3^ h^−1^ when the pit exhaust fan was operating. As expected, the inflow rate increased with an increase in the fan operation rate. However, unlike the ventilation rate, the inflow rate only changed a little at low fan operation rates and was followed by a steep increase beyond a certain fan operation rate. In pig room A, the inflow rate through the four baffles of the side exhaust fan started to increase significantly when the fan was operated at 70% or higher. For the pit exhaust fan, this trend was observed at 50% or higher. In pig room B, the inflow rate through the side and chimney exhaust fans started to increase significantly at 70% and 60% or higher, respectively.

The baffle system, which is part of the side inlet, opens due to the negative pressure generated inside by the exhaust fan. At very low negative pressures, the baffle remains closed and only opens when the negative pressure reaches a certain threshold. The threshold and degree of opening can vary depending on the strength of the spring and friction. During the experiment, it was visually observed that the baffle did not open significantly when the fan operation rate was below 50%; however, the opening of the baffle became visible when the fan operation rate reached 70% or higher.

Although the inflow rates varied slightly among the four baffles in both barns when the side exhaust fan was operating, no significant difference was observed. However, in the case of the pit exhaust fan in pig room A, there was a notable difference in the inflow rate among the baffles, even though the inflow rates themselves were very small. This difference could be attributed to the uneven operation of the baffles at very low ventilation rates. Additionally, in pig room B, the chimney exhaust fan was positioned slightly off-center near S3 (see Figure 1), which resulted in a slightly higher inflow rate at S3 compared to that at other baffles. However, overall, the difference in the inflow rate through the four baffles seemed to be influenced more by the individual mechanical performance of each baffle system rather than the shape and location of the outlet, likely due to the relatively small size of the pig rooms.

The inflow rate through the ceiling inlet ranged from 1994 to 4304 m^3^ h^−1^, except for the very low value of 311 m^3^ h^−1^ when the pit exhaust fan was operating. Compared to the side inlet, the ceiling inlet exhibited higher inflow rates. In the case of the side exhaust fan, it was not clearly discernible at which threshold fan operation rate the baffles of the ceiling inlet started to operate. This indicates that even at low ventilation rates, the baffles opened properly. Consequently, it could be observed that the inflow rate through the ceiling inlet was significantly higher compared to the side inlet across all ventilation rate ranges. When the chimney exhaust fan was used in pig room B, the inflow rate started to increase from the 50% fan operation rate onwards. In pig room A, when the pit exhaust fan was used, the inflow rate tended to increase when the fan operated at 70% or higher. The inflow rates among the eight baffles of the ceiling inlet exhibited little difference, indicating the better spatial uniformity of the inflow rates compared to that of the side inlet.

The total inflow rate through the duct inlet, except for when the pit exhaust fan was used and a low value of 608 m^3^ h^−1^ was observed, ranged from 1895 to 1999 m^3^ h^−1^, which remained relatively consistent regardless of the type of pig room or outlet exhaust fan. Among the 12 holes along the duct at different distances (24 holes in total, considering both sides), the inflow rate was significantly higher at d1, d2, and d3 (see Figure 1), which are closer to the side exhaust fan or chimney exhaust fan, compared to d7, which is closer to the corridor. In most cases, d7 exhibited an inflow rate of 15 m^3^ h^−1^ or less, and d1-d3 demonstrated higher inflow rates ranging from 100 to 157 m^3^ h^−1^. The highest inflow rates were observed at the holes closest to the exhaust fan: positions d2 and d3 for the side exhaust fan, position d3 for the chimney exhaust fan, and position d1 for the pit exhaust fan.

The occurrence of significantly higher inflow rates at holes closer to the exhaust fan indicates that the fresh air entering through these holes is more likely to be immediately expelled by the exhaust fan rather than contributing to the overall ventilation effect inside the pig room. Therefore, to improve the ventilation performance, the inflow rate differences among the holes formed in the duct should be reduced. This might require adjustments in the spacing of the holes formed in the duct.

#### 3.1.3. Infiltration Rate Estimation

The total inflow rate, which was measured based on different combinations of inlets and outlets, was significantly lower compared to the ventilation rate, indicating a considerable amount of infiltration and, thus, excessive air leakage. However, the accuracy of measuring the total inflow rate could be affected by uncertainties associated with the single-point measurement using thermal anemometers, and the estimation of the infiltration rates could be subject to these uncertainties.

Figure 10 illustrates the ratio of the infiltration rate to the actual ventilation rate. Except for the cases of using the ceiling inlet and the side inlet/side outlet configurations in pig room B, the infiltration ratios were relatively consistent across different fan operation rates. In some cases, as the fan operation rate increased, the infiltration ratio slightly decreased.

Several characteristics of the inlet and exhaust configurations in the target facility can be summarized based on the results of the infiltration ratio. First, when the fan operation rate was below approximately 60% and the ventilation rate was low, the infiltration ratio was higher. This could be attributed to the poor operation of the baffle inlet under low negative pressure conditions, which allowed for more air infiltration. Second, overall, the infiltration ratios were higher for the side inlet and ceiling inlet compared to that for the duct inlet. This is because the duct inlet is constantly kept open, whereas the side inlet and ceiling inlet rely on the improper operation of the baffles, which may result in limited air inflow. Therefore, measures such as adjusting the spring strength of the baffle inlets are necessary. Third, the infiltration ratios were considerably high overall. When examining cases with a fan operation rate of 70% or higher, the infiltration ratios in pig room A ranged from 57% to 90%, whereas in pig room B, they ranged from 59% to 78%, excluding exceptional cases. According to the literature [21], the infiltration rate in swine facilities has been reported to be 6.43 ± 1.68 ACH, with some studies [22,23] indicating that it can account for approximately 50% of the minimum ventilation rate. When dividing the calculated infiltration rates by the room volume, pig room A exhibited a range of 0.87–13.53 ACH, and pig room B showed a range of 11.64–29.19 ACH under 100% fan operation rate conditions. The high infiltration observed in the target facility could be attributed to the total open area of the inlets being very small compared to the high ventilation rates, highlighting the need for structural improvements in the inlets.

Infiltration mostly occurs through gaps in the worker entrance door, unused inlets or exhaust fans, and potentially through unsealed areas of the walls or joints of the building [24]. Especially in the research-purpose facility, which includes various inlet and exhaust combinations to allow for different ventilation scenarios, infiltration is significant. As infiltration hinders a uniform livestock environment through proper ventilation, measures to suppress the occurrence of infiltration in the research facility are required.

### 3.2. Ventilation Effectiveness Using Tracer Gas Decay Method

#### 3.2.1. Distribution of Local ACH in Pig Activity Area

Figure 11 presents the distribution of local ACH measured at the animal height of 0.3 m in pig room A, whereas Figure 12 represents the same for pig room B. The local ACH values were obtained through tracer gas decay experiments. The designed ACH values were calculated by dividing the two ventilation rates applied (2000 m^3^ h^−1^ and 5000 m^3^ h^−1^) by the room volume, which was determined to be 13.6 ACH and 33.9 ACH, respectively.

Overall, pig room B exhibited higher ACH values compared to pig room A, indicating better air exchange within this space. In both rooms, the combination of the ceiling inlet and side exhaust fan resulted in the most effective ventilation. Across all inlet and exhaust configurations, the middle locations generally exhibited the highest ventilation effectiveness.

Figure 12 shows that the use of the side exhaust fan yielded better ventilation effectiveness for pig room B, with higher ACHs than those of the chimney exhaust fan. This could be attributed to the measurement results obtained at a height of 0.3 m above the floor, which is close to the pigs’ respiratory level. While all the inlets were positioned at a higher level than the pigs’ activity area, exhausts through the ceiling, as in the case of the chimney exhaust fan, might not allow for sufficient airflow to pass through the pigs’ activity area before being exhausted. On the other hand, the side exhaust fan, which operated at a lower height through the sidewall, provided a more advantageous path for the airflow to reach the pigs’ activity area.

In both pig room A and pig room B, the use of the side exhaust fan combined with the ceiling inlet resulted in the highest ACH values, indicating the best ventilation effectiveness. The next best results were observed with the use of a duct inlet. Regarding the uniformity of ventilation effectiveness at the nine measurement points, the spatial uniformity of ventilation effectiveness was higher with the use of ceiling and duct inlets compared to the side inlet. This could be attributed to the larger number of inlets and their more spatially uniform distribution for the ceiling and duct inlets. The measurements of inflow rates at these inlets also demonstrated that the spatial distribution of inflow rates was superior for the ceiling inlet, aligning with the results of local ACH measurements. In the case of the duct inlet, the uniformity of ventilation effectiveness in pig room B was high. However, in pig room A with the duct inlet, higher ventilation effectiveness was generally observed toward the side exhaust fan side rather than the corridor side. This could be explained by the measurement results of inflow rates, which indicated that the holes closer to the exhaust fan had higher airflow rates. When using the side inlet, higher ACH values were observed near the exhaust fan rather than near the side inlet. This could be attributed to the formation of jets as the air entered through the side baffle openings in the corridor, resulting in the air not reaching the animals directly below the side inlet but moving further toward the exhaust fan before descending to reach the animal height.

In the case of using a chimney exhaust fan (Figure 12), as mentioned earlier, the overall ventilation effectiveness at animal height was lower than that at the side exhaust fan. Regarding the inlet types, both the ceiling and duct inlets exhibited similar ACH values and good uniformity. However, the ACH values were lower near the corridor side for the duct inlet, as the airflow rate was lower at the holes on the corridor side. Similarly, for the side inlet, higher ACH values were observed near where the chimney exhaust fan was positioned rather than at the corridor side.

For all ventilation configurations, the ventilation effectiveness significantly decreased when the ventilation rate was 2000 m^3^ h^−1^ compared to when it was 5000 m^3^ h^−1^. Particularly, when using the chimney exhaust fan, the exhaust location was positioned high on the ceiling relative to the animal’s activity height, resulting in poor air circulation at lower ventilation rates in the animal’s activity space. From this perspective, the combination of the chimney exhaust fan and ceiling inlet exhibited the poorest ventilation effectiveness in the animal’s activity space at lower ventilation rates. In the remaining cases, particularly when the side exhaust fan was used, the difference in ACH values among the different inlet types at lower ventilation rates was not significant. This could be attributed to the higher infiltration ratio in both pig rooms at lower ventilation rates. The choice of the inlet type had a less pronounced impact on ventilation effectiveness at lower ventilation rates. However, when the duct inlet was used, slightly higher ACH values were observed compared to the other inlet types because the duct inlet had lower infiltration than the other inlet types.

Pig room A and pig room B had the same size and shape, and when applying the side exhaust fan, they had the same ventilation configuration. However, the difference in ventilation effectiveness differed significantly between the two rooms. Overall, pig room B exhibited greater ventilation effectiveness, which could be attributed to the infiltration ratio in pig room A being higher than that in pig room B. Pig room A had a pit ventilation system, and the floor was entirely slatted, unlike pig room B which had a partially slatted floor. Additionally, pig room A had a variable structure that could be connected to the adjacent room, which was likely to have a significant impact on infiltration. Due to the higher infiltration, the influence of the inlet on ventilation effectiveness was weakened, resulting in lower ventilation efficiency in pig room A.

#### 3.2.2. Comparison of Ventilation Effectiveness between Pig and Worker Activity Spaces

To analyze the ventilation effectiveness in the pig and worker activity spaces, the results measured at two different heights are presented in Table 3, Table 4 and Table 5. The average ACH values in the pig activity space were calculated by averaging the measurements from six locations within the four pens where the pigs were housed out of the nine measurement points.

In the case of pig room A, the ACH values in the worker activity space were consistently higher than those in the pig activity space for all types of inlets. At a low ventilation rate of 2000 m^3^ h^−1^, the ACH values in the worker activity space were 1.4–3.7 times higher, and at high ventilation rate of 5000 m^3^ h^−1^, they were 1.1–1.4 times higher. This could be attributed to the fact that all inlet heights were closer to the worker activity space than the pig activity space.

In the case of pig room B, the difference in ventilation effectiveness between the pig and worker activity spaces was not significant when using the side exhaust fan. As mentioned in Section 3.2.1, this combination demonstrated overall good ventilation effectiveness, resulting in a less pronounced difference in ventilation effectiveness between the animal and worker heights. However, as shown in Table 4, when the chimney exhaust fan was used in pig room B, except for the case of using the ceiling inlet at the high ventilation rate, the difference in ventilation effectiveness between the two activity spaces ranged up to a maximum of 1.6 times.

Overall, due to the small size of the pig rooms, there was no significant difference in ventilation effectiveness between the pig and worker activity spaces. However, it is noteworthy that for inlet and exhaust combinations with good ventilation effectiveness, the difference in ventilation effectiveness between the pig and worker activity spaces was smaller. On the other hand, for ventilation configurations with poor ventilation effectiveness, the ventilation effectiveness in the pig activity space was lower compared to the worker space. Moreover, in pig room A, which had higher levels of infiltration, the ventilation effectiveness in the pig activity space was consistently lower than that in the worker activity space, regardless of the ventilation configuration. Therefore, in larger-scale commercial farms, the selection of appropriate ventilation configurations becomes more important to provide a favorable environment for both the pigs and workers.

## 4. Conclusions

This study investigated the ventilation performance in two research-specialized pig facilities (pig room A and pig room B) by examining various combinations of inlet and exhaust configurations. The key findings of this research can be summarized as follows:

(1)The maximum ventilation rates in pig room A and pig room B were measured as 6109 m^3^ h^−1^ and 7710 m^3^ h^−1^, respectively. The configurations that combined the side inlet/chimney outlet, ceiling inlet/side outlet, and duct inlet/pit outlet showed higher ventilation rates.(2)The measured maximum ventilation rates of the side, chimney, and pit exhaust fans in the experimental rooms were significantly lower (51.9%, 92.4%, and 33.6%, respectively) than the rates specified by the manufacturer. Therefore, relying on measured ventilation rates rather than fan operation rates is essential when using these pig facilities for research purposes.(3)The ceiling inlet was found to provide better airflow uniformity at individual inlet locations compared to the side inlet. The duct inlet had significant differences in airflow rates among individual holes along the length of the duct, with a substantial increase in the airflow near the exhaust fan.(4)The total inflow rates were much lower than the ventilation rates in both facilities. It was estimated that pig room A had infiltration rates ranging from 57% to 90% of the ventilation rate, while pig room B had infiltration rates ranging from 59% to 78%. High infiltration rates can reduce the influence of ventilation configurations and hinder ventilation effectiveness, highlighting the need for remedial actions.(5)The use of the side exhaust fan, which was closer to the pig activity space, resulted in higher ventilation effectiveness in the pig activity area compared to using the chimney exhaust fan. The use of the ceiling inlet, rather than the side inlet, not only improved ventilation effectiveness but also provided better spatial uniformity. In the case of using the side inlet, the jet effect of inflowing air through the baffle openings resulted in lower ventilation effectiveness in the pig area near the side inlet.(6)The ceiling inlet and side outlet combination exhibited the highest air exchange rates at a ventilation rate of 5000 m^3^ h^−1^. In general, the ventilation effectiveness significantly decreased when the ventilation rate was 2000 m^3^ h^−1^ compared to when it was 5000 m^3^ h^−1^ for all inlet and exhaust combinations.(7)In most combinations of inlet and exhaust configurations, the worker activity space exhibited higher air exchange rates than the pig activity space. However, in the small-scale facilities examined in this study, the difference was not substantial. Nevertheless, this difference could be more pronounced in ventilation configurations with lower effectiveness and facilities with higher infiltration ratios.

It is important to note that the conclusions drawn in this study are based on the assessment of ventilation rates and efficiency in the absence of animals and workers. In reality, the presence of these biological systems may influence ventilation rates and efficiency. Nonetheless, this research highlights the importance of selecting appropriate inlet and exhaust configurations. By considering factors such as ventilation rates, airflow distribution, and infiltration control, optimal ventilation systems can be designed to create a comfortable and healthy environment for both pigs and workers in pig production facilities.

## Figures and Tables

**Figure 1 animals-13-02451-f001:**
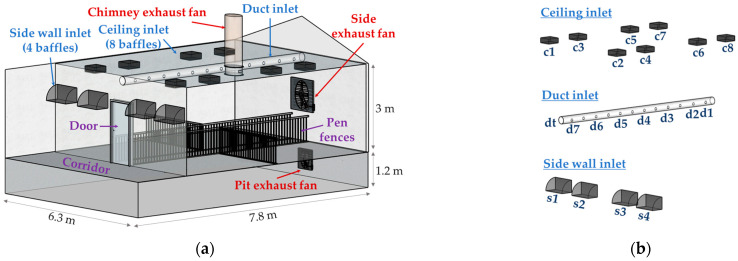
Schematic diagram of (**a**) Ventilation configurations of the experimental pig room and (**b**) Anemometer positions for inflow rate measurements. The drawing was created using SketchUp (Trimble Inc., Sunnyvale, CA, USA).

**Figure 2 animals-13-02451-f002:**
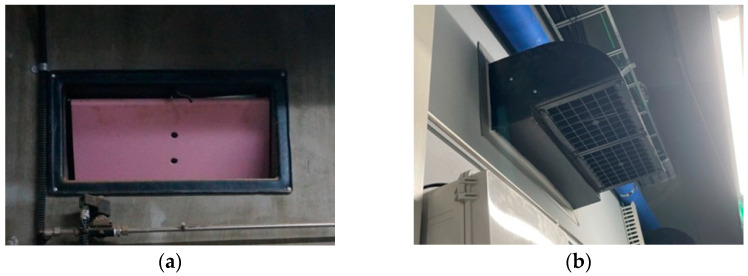
Configuration of the side baffle inlet from (**a**) Indoor side and (**b**) Corridor side.

**Figure 3 animals-13-02451-f003:**
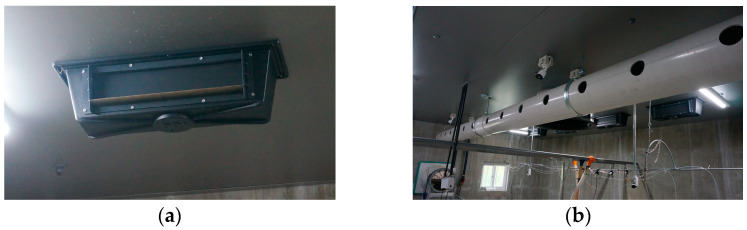
Configuration of the (**a**) Ceiling baffle inlet and (**b**) Duct inlet.

**Figure 4 animals-13-02451-f004:**
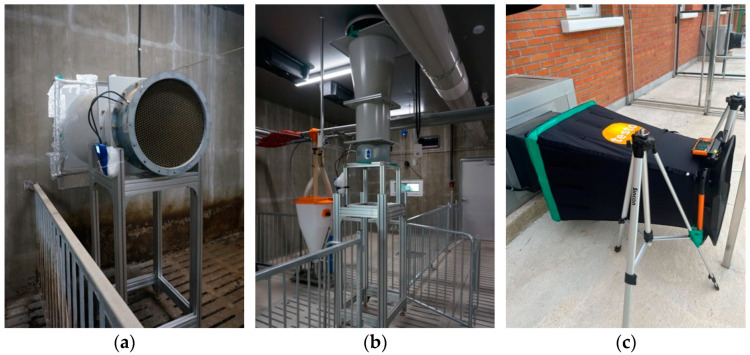
Measurement of actual ventilation rates of exhaust fans using hood-type anemometers. (**a**) Side exhaust fan, (**b**) Chimney fan, and (**c**) Pit exhaust fan.

**Figure 5 animals-13-02451-f005:**
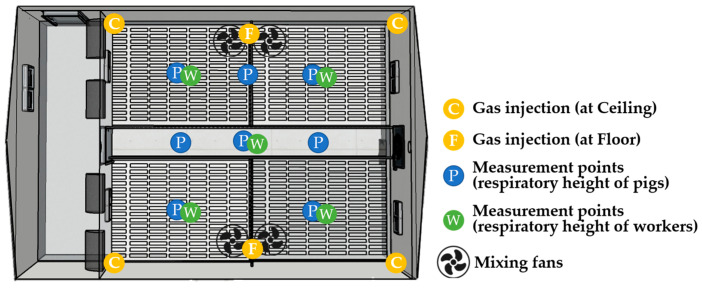
Layout of the tracer gas experiment, including gas injection, mixing fans, and gas measurement locations.

**Figure 6 animals-13-02451-f006:**
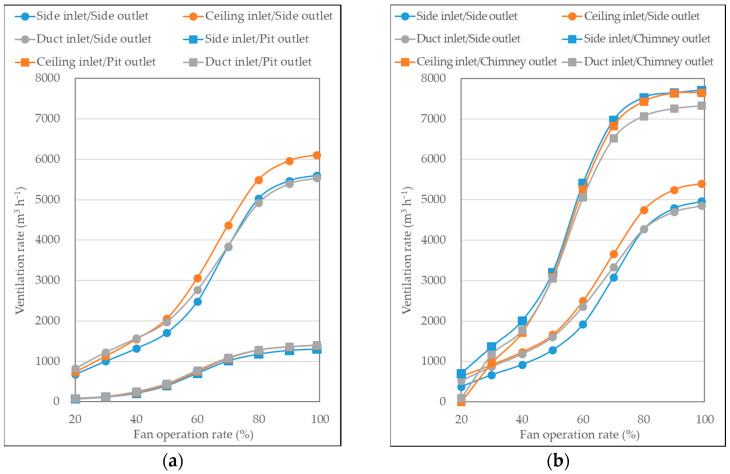
Actual ventilation rates according to six ventilation configurations in (**a**) Pig room A and (**b**) Pig room B.

**Figure 7 animals-13-02451-f007:**
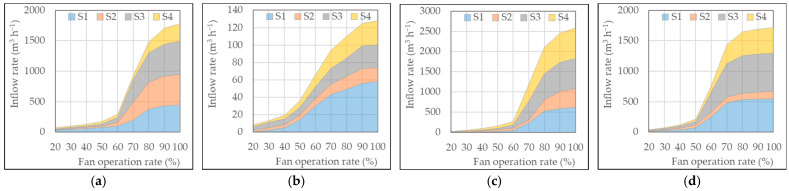
Inflow rates through four baffles of the side inlet according to different outlet configurations: (**a**) Pig room A with side exhaust fan, (**b**) Pig room A with pit exhaust fan, (**c**) Pig room B with side exhaust fan, and (**d**) Pig room B with chimney exhaust fan.

**Figure 8 animals-13-02451-f008:**
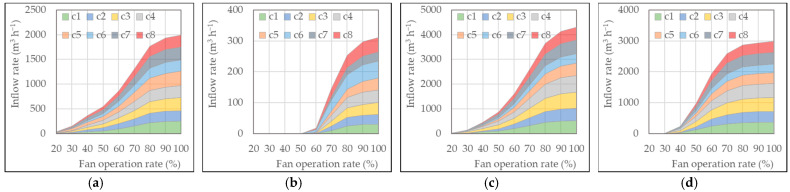
Inflow rates through eight baffles of the ceiling inlet for different outlet configurations: (**a**) Pig room A with side exhaust fan, (**b**) Pig room A with pit exhaust fan, (**c**) Pig room B with side exhaust fan, and (**d**) Pig room B with chimney exhaust fan.

**Figure 9 animals-13-02451-f009:**
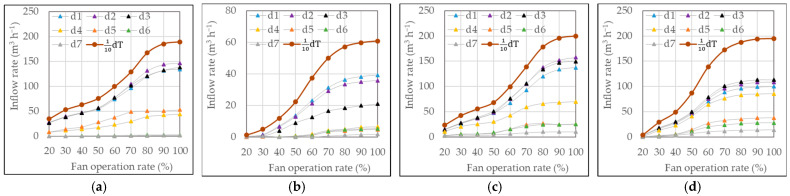
Inflow rates through seven holes and the corridor-side opening (dT) of the duct inlet for different outlet configurations: (**a**) Pig room A with side exhaust fan, (**b**) Pig room A with pit exhaust fan, (**c**) Pig room B with side exhaust fan, and (**d**) Pig room B with chimney exhaust fan. dT is equal to 10 times the value of $\frac{1}{10}$ dT.

**Figure 10 animals-13-02451-f010:**
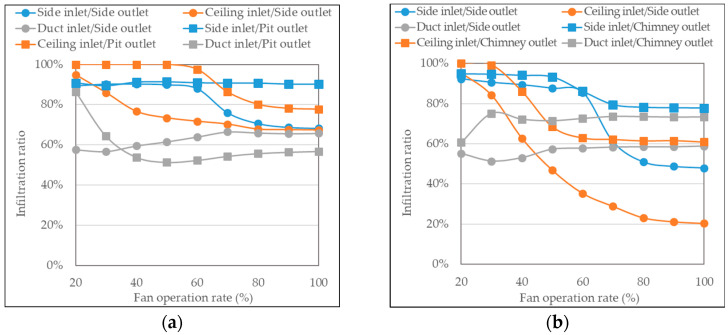
Ratio of infiltration rate to the actual ventilation rates according to six ventilation configurations in (**a**) Pig room A and (**b**) Pig room B.

**Figure 11 animals-13-02451-f011:**
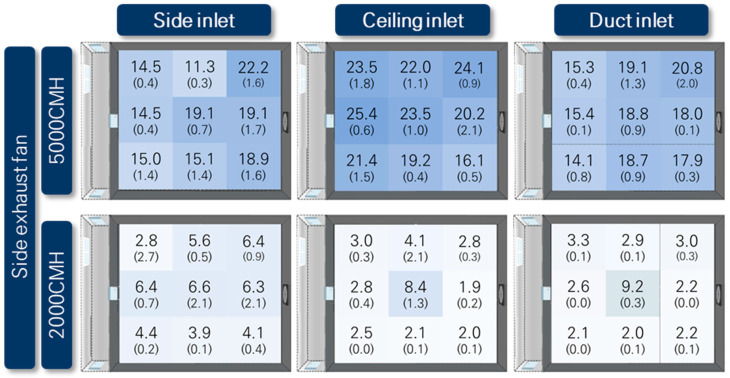
Spatial distribution of air changes per hour (ACH) in Pig Room A based on tracer gas decay experiments. The numbers in parentheses represent the standard deviations from three repeated experiments. CMH stands for cubic meters per hour.

**Figure 12 animals-13-02451-f012:**
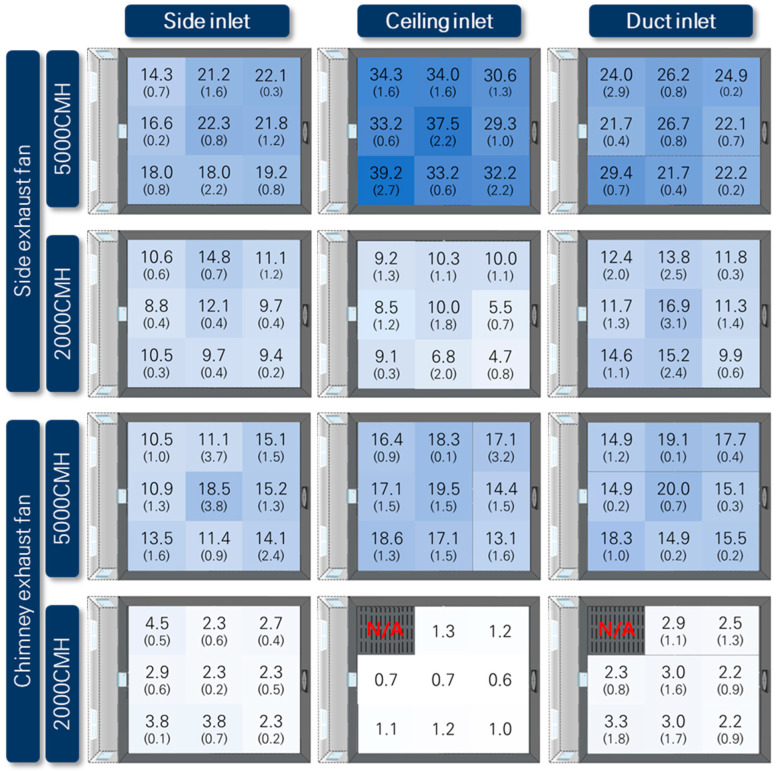
Spatial distribution of air changes per hour (ACH) in Pig Room B based on tracer gas decay experiments. The numbers in parentheses represent the standard deviations from three repeated experiments. CMH stands for cubic meters per hour. “N/A” indicates values that were not properly measured due to errors in the gas measurement device.

**Table 1 animals-13-02451-t001:** Experimental cases and conditions for measuring the ventilation rate and air changes per hour (ACH).

Conditions		Pig Room A	Pig Room B
Air inlets		Side, Ceiling, Duct	
Exhaust fans		Side, Pit ^1^	Side Chimney
Fan operation rates for measuring	Ventilation rate	10, 20, 30, 40, 50, 60, 70, 80, 90, 100%	
	ACH	Adjusted for ventilation rates of 2000 m^3^ h^−1^ and 5000 m^3^ h^−1^	

^1^ Pit exhaust fan was not used for ACH measurements.

**Table 2 animals-13-02451-t002:** Experimental periods and average air temperature and humidity during the experiments.

	Conditions	Periods	Air Temperature (°C)		Humidity (%)	
Experiments			Indoor	Outdoor	Indoor	Outdoor
Pig room A, VR ^1^		9–13 May 2022	19.2	18.8	69.0	59.0
Pig room A, ACH ^2^		16–20 May 2022	20.3	18.9	49.0	42.9
Pig room B, VR ^1^		11–22 July 2022	26.5	25.6	83.1	74.7
Pig room B, ACH ^2^		25–29 July 2022	28.5	28.3	74.6	71.9

^1^ VR: ventilation rate measurement; ^2^ ACH: air changes per rate of measurement.

**Table 3 animals-13-02451-t003:** Air changes per hour (ACH) within pig and worker activity spaces in pig room A with side exhaust fan.

	Ventilation Rate Inlet Types	2000 m^3^ h^−1^			5000 m^3^ h^−1^		
Target Spaces		Side	Ceiling	Duct	Side	Ceiling	Duct
Pigs		4.5 ± 0.8	2.7 ± 0.5	2.6 ± 0.1	15.5 ± 1.0	21.1 ± 1.0	17.5 ± 1.1
Workers		6.3 ± 2.0	8.4 ± 1.3	9.6 ± 2.3	21.1 ± 2.2	23.4 ± 2.8	19.8 ± 2.5

**Table 4 animals-13-02451-t004:** Air changes per hour (ACH) within pig and worker activity spaces in pig room B with side exhaust fan.

	Ventilation Rate Inlet Types	2000 m^3^ h^−1^			5000 m^3^ h^−1^		
Target Spaces		Side	Ceiling	Duct	Side	Ceiling	Duct
Pigs		11.0 ± 0.5	8.4 ± 1.1	12.9 ± 1.5	18.8 ± 1.1	33.9 ± 1.7	24.7 ± 0.9
Workers		10.1 ± 1.1	7.7 ± 1.8	10.2 ± 1.4	20.3 ± 4.7	32.0 ± 4.3	24.5 ± 2.7

**Table 5 animals-13-02451-t005:** Air changes per hour (ACH) within pig and worker activity spaces in pig room B with chimney exhaust fan.

	Ventilation Rate Inlet Types	2000 m^3^ h^−1^			5000 m^3^ h^−1^		
Target Spaces		Side	Ceiling	Duct	Side	Ceiling	Duct
Pigs		3.1 ± 0.3	1.1 ± 1.1	2.8 ± 1.4	12.6 ± 1.7	16.8 ± 1.4	16.7 ± 0.5
Workers		3.4 ± 0.7	1.8 ± 0.4	3.4 ± 1.0	16.1 ± 4.8	15.2 ± 1.6	17.2 ± 2.0

## Data Availability

The data presented in this study are available on request from the corresponding author. The data are not publicly available due to privacy concerns.

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
