# Peer review of "Evaluation of Actual Ventilation Rates and Efficiency in Research-Scale Pig Houses Based on Ventilation Configurations"

_animals, 2023, doi:10.3390/ani13152451_

Round 1
Reviewer 1 Report
1. The language should be improved.
2 . Line 59-60 is too general.
3. Fig.1 adds the name of the modeling software.
4. What is the evaluation index of uniformity of flow.
5.How to consider the change of flow caused by air density?
6.What are the advantages of the method in this study compared to CFD and why CFD was not used ?
Some words need further correction
Author Response
Thank you for your review comments and suggestions. We greatly appreciate your feedback. Based on your comments, we have made revisions to the manuscript. Please refer to the attached file for the specific details.

Reviewer 2 Report
The manuscript is interesting and publishable, but the main issue I have is about the experimental design and statistic analysis. It is my understanding that this research is not observational, therefore, please specifically define the experimental design and the statistic analysis method under such design. Please also show your statistic test results to support your discussion
The English writing is acceptable
Author Response

(The authors gave the same response as above.)

Reviewer 3 Report
Regardless of the formulation of the general purpose of the research, I suggest that the authors write what was the cognitive (scientific) goal and what was the utilitarian (useful) goal. If a scientific and useful goal is clearly defined, then in the Conclusions chapter it can be referred to and it can be stated whether the set goals have been achieved and what are the prospects (scope) of further research to be carried out in the subject area under consideration. In the last paragraph of the Introduction chapter, the authors wrote about creating a solid scientific basis for experiments in the future, but it would be worth being more specific about what exactly was the scientific goal of the study, as well as the utilitarian goal.
Before stating the purpose of the research study, it would be worth formulating the research problem. I think that based on the review of the state of knowledge presented in the Introduction of the article, one can easily formulate a research problem. I suggest that in the summary of the state of the art simply write the sentence: "The research problem is ...". The research problem can be linked to the presentation of a gap in the current state of knowledge, which is then translated into the formulation of the research goal / study.
In the Abstract, the authors wrote that they tested the effectiveness of ventilation. I think that it would be worth adding information about what the authors understood by the term "effectiveness". While the wording ventilation rates (line: 28) is unambiguous, the concept of ventilation efficiency (line: 29) requires a more precise interpretation. Efficiency is a concept that requires more detail to know what exactly efficiency was when reading the Abstract. Anyway, in the article itself, it would be worth specifying exactly what is included in the concept of efficiency, what indicators are taken into account in the efficiency of ventilation.
I would like to know how the experiment performed and its results would compare to real conditions when there are pigs in the pens. Could the presence of animals in the pens affect the results / indicators in the studies? After all, animals emit heat and gases (as a result of the work of the digestive system), change their position in the pen, raise dust from the bedding on the substrate (floor), which has an impact on the microclimate in the livestock room for pigs. In real conditions, the density of animals in the pen is certainly also important. How could these factors affect the measurement results obtained? Perhaps it is worth mentioning some correction factors here, which would relate the research results to real conditions, taking into account pens inhabited by pigs.
Can the share of dust and other such pollutants in the air in the building clog the openings through which the air flows, reducing their throughput? Was air pollution with dust included in the experiment?
I would like to ask for what number of animals in the building / herd the ventilation systems included in the study were designed? It is worth writing about it in the article.
In the caption to Figure 7, it would be worth specifying what S1, S2, S3 and S4 mean, which are on the individual Figures.
Figures 11 and 12 show the parameters 2000 CMH and 5000 CMH. It would be worth explaining in the figure captions what CMH means. I can guess that it's about Cubic Meters per Hour, but it's worth giving this information in the article.
In the formula (1) it would be worth specifying the units for the parameters (C and t) taken into account. Only the unit for ACH is given in this formula.
I would like to know to what extent the air flow rate (in cubic meters per hour) could affect the comfort and welfare of the pigs in the pen as well as the people working in the building. Could this factor be taken into account in the analysis? The research was conducted without the participation of animals, and animals should be the main beneficiaries, just like working people, improving knowledge by the example of ventilation assessment and its design in pig housing.
I don't quite understand the construction of Table 1. In the first main column (Conditions) there are two columns at the bottom. I don't know which of these columns refers to the two rows on the right side of the Table. This is unclear to me and needs correction in my opinion.
Table 2 shows data from a period when the temperature inside and outside the livestock shed is very similar. In my opinion, it would also be worth showing the results of the ventilation system when the temperatures inside and outside the livestock room are more varied. Thanks to this, a more complete picture of the effects of ventilation would be shown. Of course, this aspect may be announced as a research goal in the future.
The authors use the ACH indicator, i.e. air change per hour. It seems to me that 'air exchange per hour' would be more appropriate in the present case. This is an issue for the authors to consider.
In the part presenting the results of own research, it would be worthwhile to compare these results with the research of other authors to a greater extent, i.e. to develop a discussion of the research results.
Author Response

(The authors gave the same response as above.)
